# Comparative analysis of multiorgan toxicity induced by long term use of disease modifying anti-rheumatic drugs

**Tamseela Mumtaz**[1]*, **Ayesha Tahir**[1‡], **Maham Almas Tariq**[1‡], **Muhammad Amir Iqbal**[2]

**1** Faculty of Science and Technology, Department of Zoology, Government College Women University, Faisalabad, Punjab, Pakistan, **2** Institute of Zoology, University of the Punjab, Lahore, Punjab, Pakistan

☯ These authors contributed equally to this work.
‡ These authors also contributed equally to this work
* dr.tamseelamumtaz@gcwuf.edu.pk

**Data Availability Statement:** All relevant data have been uploaded as supporting information files.

**Funding:** The author(s) received no specific funding for this work

## Abstract

The constant use of disease modifying anti rheumatic drugs affects the functioning of multiple organs inside the body. Some drugs are more toxic than others. The present case control investigation was designed to evaluate the comparative toxicity of methotrexate and leflunomide on multiple organs in rheumatoid arthritis patients. For this purpose, 100 subjects with confirmed rheumatoid arthritis condition were recruited form tertiary care center. Whereas 50 age matched controls were recruited from the local healthy population. Participants of the study were categorized into three groups with equal numbers of subjects in each group (n = 50). Group 1 comprised rheumatoid arthritis patients on methotrexate treatment, group 2 included rheumatoid arthritis patients on leflunomide treatment and group 3 were healthy subjects. Cardiac and respiratory response was evaluated by monitoring blood pressure, pulse and breathing rate and spot oxygen saturation. Stress on liver was estimated by measuring change in liver enzymes (alanine transaminase, aspartate aminotransferase, and alkaline phosphatase) and total bilirubin. While, degree of renal impairment was assessed by calculating glomerular filtration rate, serum creatinine, urinary urea and uric acid. For statistical interpretation, data was subjected to independent student "t" test and analysis of variance (one way ANOVA) for mean variations. Both methotrexate and leflunomide elevated the systolic and diastolic blood pressure and pulse rate. Leflunomide maintained the oxygen saturation at 96.7%, whereas methotrexate exerted serious effect on spot oxygen saturation by reducing it significantly to 93.25% than healthy subjects. Hepatotoxicity manifested by sustained use of leflunomide was perceptible in this study group. Whereas, both methotrexate and leflunomide influenced renal function as indicated by marked increase in blood urea nitrogen (P = 0.001), serum creatinine (P = 0.007) and reduced glomerular filtration rate (P<0.0001). However, use of methotrexate demonstrated significant (P<0.0001) reduction in serum uric acid and urinary urea levels. Methotrexate is more injurious to heart, blood vessels and kidneys than leflunomide but it is less noxious to hepatic parenchyma. Contrarily, leflunomide usage is comparatively better option for respiratory, cardiovascular, and renal health but dangerous to liver. Thus, a single drug can't be

**Competing interests:** The authors have declared that no competing interests exist

**Abbreviations:** RA, Rheumatoid Arthritis; DMARDs, Disease Modifying Anti Rheumatic Drugs; MTX, Methotrexate; LEF, Leflunomide; GFR, Glomerular Filtration Rate; GN, Glomerular Nephritis; CKD, Chronic Kidney Disease; BUN, Blood Urea Nitrogen; $SPO_2$, Spot Oxygen Saturation; ALT, Alanine Transaminase; AST, Aspartate Aminotransferase; ALP, Alkaline Phosphatase; DHEA, Dehydroepiandrosterone.

prescribed for the treatment of rheumatoid arthritis for longer management of arthritis patients.

## Introduction

Rheumatoid arthritis (RA) is a persistent, integral autoimmune ailment associated with various life-threatening problems. It is an autoimmune disability of long-term joint inflammation that leads to bone destruction. It instigates in the joints, but never culminates there and mainly affects muscles, ligaments, tendons, bones, small joints of hands and feet that may lead to unrelieved pain, swelling, redness, tenderness, and potential loss of joints [1]. According to the World Health Organization (WHO), up to 14 million rheumatoid arthritis patients were reported around the world. The overall incidence of RA across the world is 0.5–1%. Rheumatoid arthritis frequently occurs at the age of 25–55 years. Rheumatoid arthritis affects over 70% of women and 55% of those over the age of 55 [2,3]. Females are two to three times more affected than males because of two reasons. First, females have stronger and more reactive immune system than men. They exhibit more T cell activation, but they are also more susceptible to developing autoimmune diseases like rheumatoid arthritis., secondly, feminine hormones like estrogens and adrenocorticosteroids (Dehydroepiandrosterone (DHEA) also flare RA risks [4].

Disease Modifying Anti Rheumatic Drugs (DMARDs) are first and foremost therapy of rheumatoid arthritis. These drugs are not pain killers but immune suppressive that slow down pain, swelling and damage of joints [5]. Methotrexate (MTX) is an antimetabolite of antifolate that suppresses immune system and used to treat rheumatoid arthritis. Leflunomide (LEF) is a new oral DMARD that acts as an immunomodulatory agent [6]. Patients treated with both MTX and LEF had been reported a significantly higher blood pressure and increased risk of CVD but the relationship between DMARDs and CVD risk is still unrevealed [7,8].

Extra articular manifestations of RA in hepatic and renal systems can be due to improper usage of DMRDs [9,10]. There is a greater risk of reduced kidney function and serious liver damage in RA patients by misuse of DMARDs. Whether or not the markers of renal disease are present, a GFR of <60 mL / min shows clear cut renal disability. Rheumatoid arthritis patients reported to have developed less than 60 mL / minute GFR as compared to general population [11].

Till now, MTX and LEF are in common practice to treat RA but both drugs have toxic effects on multiple body organs thus reducing the quality of life. Therefore, it is demand of the time to check the efficacy of these drugs against RA and toxicity to multiple organs simultaneously. Keeping in mind the above-mentioned point, this comprehensive case control study was designed to sort out the lowest destructive drug with great benefits for treatment of RA. Hence, finding the most suitable drug with minimum side effects and great efficacy to cure RA resulting in enhanced eminence of the healthy living is of huge medical importance.

## Materials and methods

### Study design

This is a case control study comprises hundred (100) subjects diagnosed with rheumatoid arthritis and fifty (50) age and sex matched healthy subjects from random population as control. The sample size was computed through G*Power software (V 3.1.9.7). The power analysis "A priori: Compute required sample size" was carried out with effect size (f = 0.40) and α =

0.05. The actual statistical power of sample size was 0.99. The study was designed to assess the risk of renal, hepatic, and cardiovascular toxicity due to long term use of DMARDs. The study plan was presented before Institutional Ethical Review Committee (IERC) of Government College Women University, Faisalabad for evaluating potential risks and benefits associated with the use of DMARDS and got approved vide letter # GCWUF/IERC/21/131. All clinical investigation was done according to the Declaration of Helsinki. The study was started in January 2019 and accomplished in August 2021.

**Sample population.** Clinically and physician diagnosed rheumatoid arthritis patients were approached from Rheumatology unit 1 of Allied Hospital Faisalabad, Punjab, Pakistan and selected by applying exclusion and inclusion criteria set for the study. The samples were chosen based on how many patients visited the health facility and how many met the criteria (S1 Fig). Inclusion criteria for case samples embrace clinical/physician diagnosed rheumatoid arthritis patients on regular administration of oral DMARDs (either MTX or LEF as monotherapy or with combination of hydroxychloroquine) since last two years. The chosen rheumatoid arthritis subjects were categorized into two subgroups; group 1: RA patients on methotrexate treatment (n = 50), group 2: RA patients on leflunomide treatment (n = 50). The age of study participants ranged from 18–65 years. Majority of the subjects had been diagnosed to have RA for last four years. All the demographic variables, disease history, extent of medicine intake and adverse outcomes of used drugs were recorded through predesigned questionnaire (S1 File). No gender restriction was applied but age was considered, and participant of less than 18 years were not included in the study. Patients with any other musculoskeletal disease (gouts, osteoarthritis, and osteoporosis) and viral infection (hepatitis B and C) were not fit for the study. Patients being irregular in follow up or showing poor consent in treatment were excluded from the analysis. Healthy, age and sex matched individuals (n = 50) from general population that neither had ailment nor any infection history were chosen as control for comparison. General medical screening was done to choose subjects as controls. The case to control ratio was set as 1:1 that means one control was recruited against one case subject of each case group. Before collecting any information from the study participants, a well explained, written, informed consent was obtained from patients and control subjects that clearly mentioned that all the gathered information strictly be used for research purpose only with maintained confidentiality. No data was pooled from archived medical records of the patients.

**Sample collection.** After fulfilling all ethical and research aspects, biophysical variables like systolic and diastolic blood pressure, pulse rate, spot oxygen saturation and body mass indices were measured. After that patients as well as healthy individuals were headed for blood sample withdrawal through venipuncture for detection of biochemical parameters. Urine samples were also collected for urine analysis.

**Serum analysis.** With respect to biochemical variants, liver enzymes (ALT, AST and GGT), were detected by UV-test recommended by IFCC (International Federation of Clinical Chemistry and Laboratory Medicine) through standard laboratory methods by means of commercially available kits (DiaSys Multipurpose kits, Cat. Nos. 127019910026, 126019910026,104419910026, respectively) at 340nm. Total, direct, and indirect bilirubin was analyzed on semi-automatic biochemistry analyzer (SA-20 CLINDIAG, Belgium) by commercially available kit (DiaSys Multipurpose kits, Cat. No 108499990336). Blood urea nitrogen and serum creatinine were detected through commercially available kits (EIABUN and EIACUN) by Invitrogen.

**Urine analysis.** Urine samples were also collected for the detection of urea, uric acid, and Glomerular filtration rate (GFR). GFR of each study participants were calculated with the help

of Cockcroft Gault formula [12] as,

$$GFR\ (mL/min) = k \times (140 - age) \times body\ weight\ /SCr(\ in\ \mu mol/L)$$

Where, k = 1.04 (female) or 1.23 (male).

**Data analysis.**   The obtained data of all variables of control and rheumatoid arthritis patients receiving varied doses of methotrexate and leflunomide were statistically analyzed by applying independent student "t" test and one way analysis of variance (one way ANOVA) on Graph Pad Prism software (version 8.3.0). The significance of obtained results was observed at P< 0.05.

# Results

Females are more vulnerable to RA than male and only 3 males were reported with RA during study period. Out of 100 study participants, 50% of the patients were on oral administration of methotrexate and other 50% patients were on leflunomide treatment with varied doses. It was observed that overall, 80% patients (n = 80) were treated with comparatively low dose (10 mg). The recommended dose was depended upon age and weight of the patients.

## Distribution of age

The average age of study participant was 45 years. In routine practice, methotrexate was advised to middle aged (31–45 years) patients at early stage of RA. While patients of 61–75 years of age that were suffering from advanced stage of ailment were prescribed with leflunomide (Fig 1).

## Side effects

Both methotrexate and leflunomide had some side effects such as fatigue, headache, alopecia, teeth pain, rash, dizziness, sleep problem, cough, depression, breathing problem and muscle stretching. Because the number of patients reporting side effects was limited, the severity of adverse effects could not be categorized into mild, moderate and severe but symptoms were relatively worse in patients treated with leflunomide (Fig 2).

## Body mass index

Both methotrexate and leflunomide drugs are responsible for weight gain due to fluid retention. BMI of MTX and LEF treated patients was greater than healthy individuals by 27.6% and 19.1%, respectively. However, gain in weight is more obvious in MTX treated patients than LEF (11.75%) (Fig 3 and Table 1).

## Blood pressure, pulse rate and spot oxygen saturation

Continuous use of DMARDs enhances the blood pressure and increase was more obvious in leflunomide treated patients. There was 16.42% increase in systolic blood pressure using leflunomide treatment. Elevation in diastolic blood pressure is almost the same for both MTX and LEF (8.6% and 8.9%, respectively). MTX treated patients have 14.6% higher pulse rate than control (P<0.0001). LEF treatment also speeds up the pulse rate (10.8%) and this elevation is statistically significant (P<0.05). Spot oxygen saturation drastically reduced (3.4%) in MTX users as compared to healthy subject. No effect on oxygen saturation was seen by LEF treatment (Table 1).

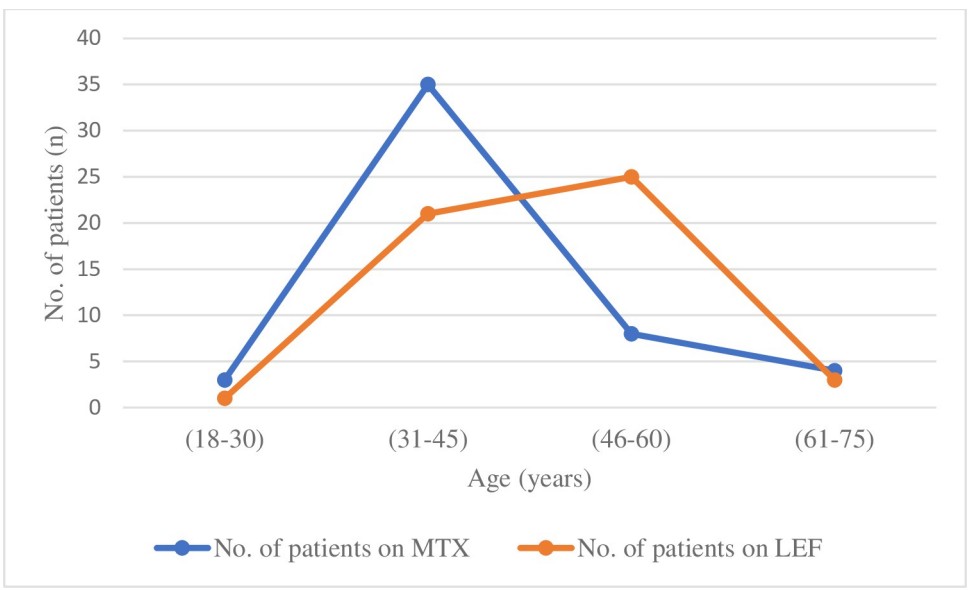

**Fig 1. Bimodal distribution of age according to prescribed treatment.** Patients of 31–45 years of age were mostly treated with MTX, while patients>45 years were treated with LEF.

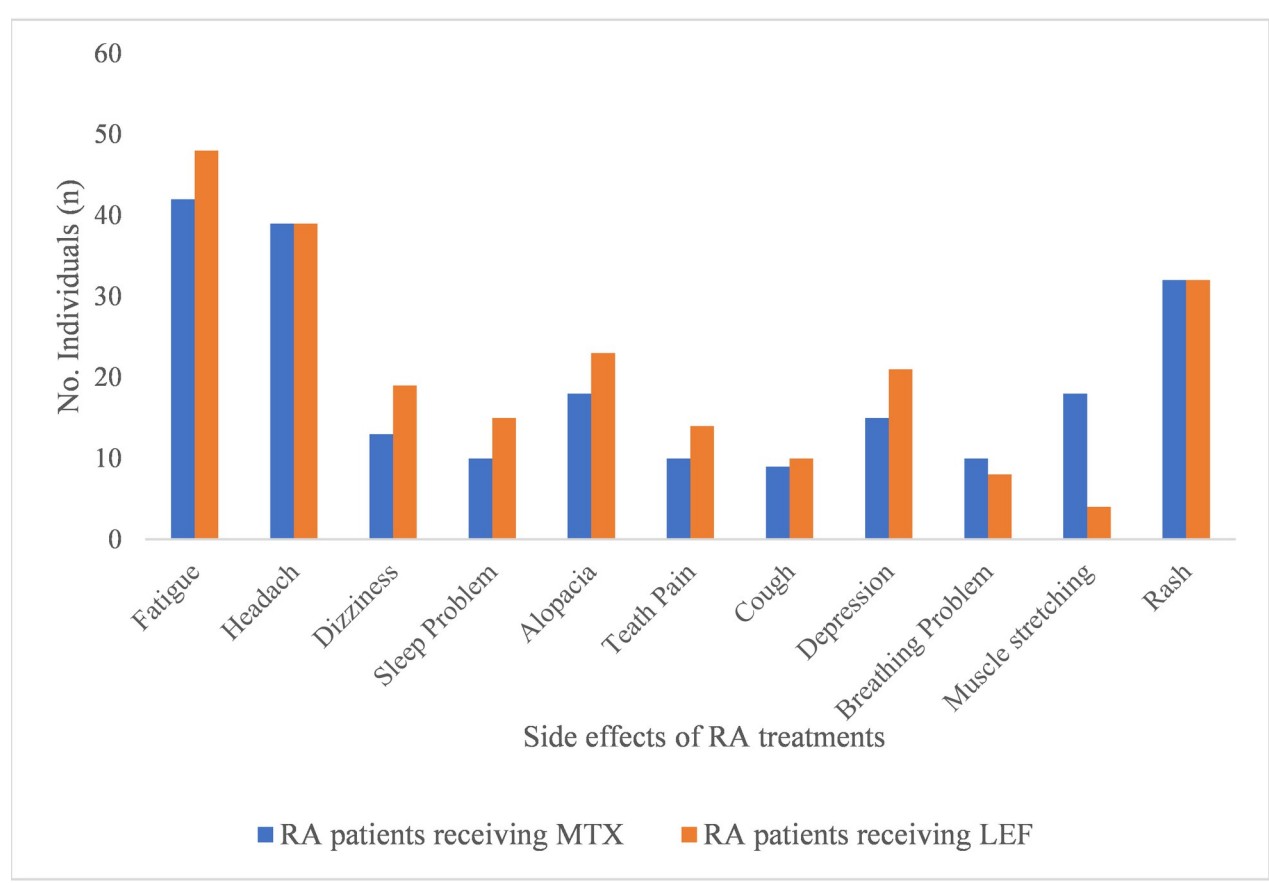

**Fig 2. Side effects exhibited by patients after methotrexate and leflunomide treatment.** Fatigue, headache and body rashes are more prominent side effects of RA treatments.

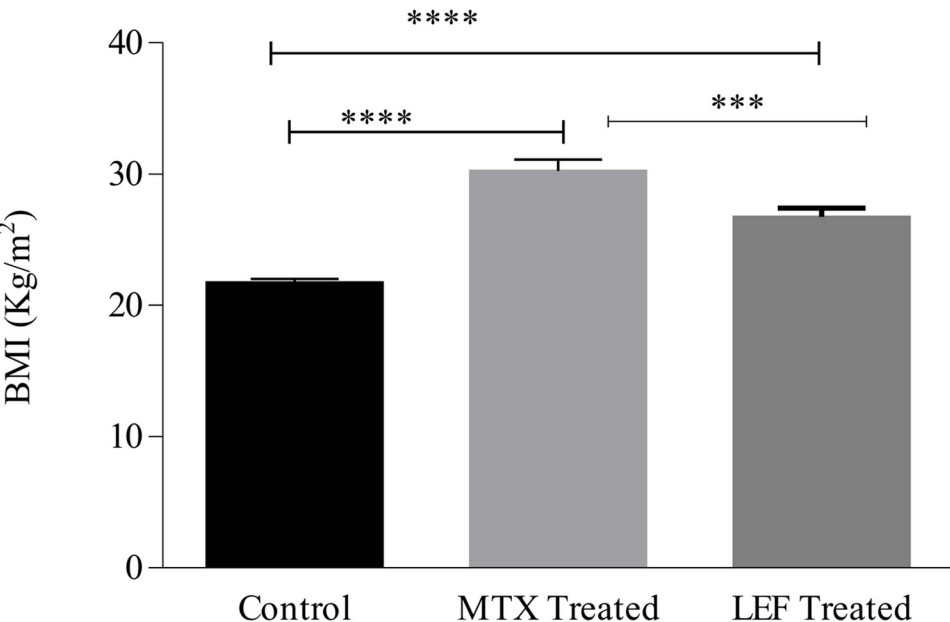

**Fig 3. Comparison of weight gain in leflunomide and methotrexate treated patients with control.** ***indicate significance at P<0.0001.

Dose adjustment affects the blood pressure and pulse rate. Low dose (>10 mg) of MTX and LEF raised the systolic and diastolic blood pressure, but the elevation varied individually. Higher dose (>20 mg), however, depicted no effect on blood pressure.

Comparison showed that LEF at low concentration ($\geq$10 mg) is safe for the heart as compared to MTX at the same concentration, (Fig 4A). Leflunomide at low concentration ($\geq$10 mg) was also safe for lungs as compared to methotrexate at the same dose. Higher dose of methotrexate ($\geq$20 mg) put serious effect on $SPO_2$, and it became lower than 90% (P = 0.009). Leflunomide even at higher dose was safe for $SPO_2$ and it maintained $SPO_2$ above 90% (P = 0.01) (Fig 4B).

**Table 1. Percentage variations in BMI, blood pressure, spot oxygen saturation, and pulse rate in healthy individuals, methotrexate and leflunomide treated rheumatoid arthritis patients.**

| Groups | Variable | Mean+SEM | Percentage variation | P value |
|---|---|---|---|---|
| **Control** | **BMI** | 21.75 ± 0.90 | | |
| **MTX treated** | **Blood pressure** | 112.0 ± 1.55 | 39.12% | <0.0001 |
| **LEF treated** | Systolic | 73.80 ± 1.31 | 12.85% | <0.0001 |
| | Diastolic | 96.50 ± 0.35 | 8.67% | <0.001 |
| | **$SPO_2$** | 86.12 ± 2.18 | 3.4% | <0.01 |
| | **Pulse Rate** | 30.26 ± 0.90 | 14.6% | <0.0001 |
| | **BMI** | 126.4 ± 2.08 | 23.03% | <0.0001 |
| | **Blood pressure** | 80.20 ± 1.53 | 16.42% | <0.0001 |
| | Systolic | 93.25 ± 0.98 | 8.94% | <0.0001 |
| | Diastolic | 98.7 ± 3.07 | 0.2% | NS |
| | **$SPO_2$** | 26.76 ± 0.90 | 10.8% | <0.05 |
| | **Pulse Rate** | 130.4 ± 2.42 | | |
| | **BMI** | 80.40 ± 2.11 | | |
| | **Blood pressure** | 96.70 ± 0.54 | | |
| | Systolic | 95.46 ± 3.05 | | |
| | Diastolic | | | |
| | **$SPO_2$** | | | |
| | **Pulse Rate** | | | |

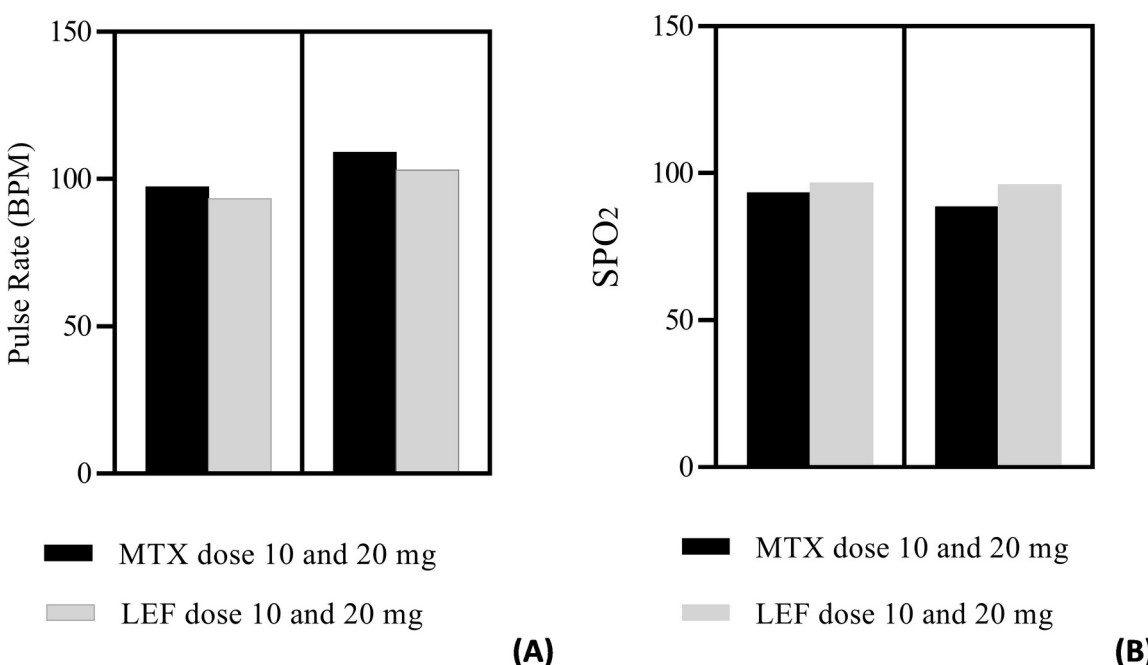

**Fig 4.** Pulse rate (BPM) (A) and Spot oxygen saturation (B) in rheumatoid arthritis patients on >10 mg and >20 mg dose of methotrexate and leflunomide.

**Liver function analysis.** MTX and LEF treatment raised ALT by 3 folds than control subjects. The levels of ALT in MTX and LEF treated groups were found to be 42.68 ± 2.14 U/L and 43.14±2.14 U/L respectively, that was 158.8% and 147.05% higher than healthy subjects. Similarly, both MTX and LEF raised AST, but LEF treatment leaves more lethal impact than MTX. The mean level of AST in LEF treated group was 43.52 ± 1.92 U/L. That was 65.09% higher than healthy subjects, whereas, MTX raised AST level up to 58.8% with an average level of 41.86 ± 1.92 U/L.

Similar trend was observed in ALP where methotrexate and leflunomide treated patients had higher ALP level than control group, and this increase was statistically significant in LEF treated patients (19.1%). The level of total, direct and indirect bilirubin was found to be lower in diseased groups than normal. Long term use of DMARDs maintained the level of circulating total bilirubin near to normal and reduced only 1.87% by MTX and 1.47% by LEF, as compared to control. The reduction was obvious in unconjugated (indirect) bilirubin by 3.8%. The effect of MTX is quite surprising as it increases the level of indirect bilirubin up to 14.4%. On the contrary, MTX reduced conjugated bilirubin by 14.7% as compared to LEF, where reduction was only 8.13% (Table 2 and Fig 5A–5F).

**Renal function analysis.** There was elevation noticed in the BUN in both groups having DMARDS doses. However, statistically significant (P = 0.0011) elevation of BUN was evidenced in subjects having MTX treatment. The elevation in BUN values were 76% and 39% higher than control Hence, MTX is more dangerous for liver than LEF. Serum creatinine levels were also affected by long term administration of DMARDs and almost doubled showing 88% increase than healthy individuals by using MTX. Whereas LEF is less toxic for kidneys than MTX. Serum creatinine level is inversely proportional to Glomerular filtration rate so an increase in serum creatinine level reduced the filtration rate. As MTX shoots up serum creatinine, it markedly suppressed GFR by 29.6% as compared to healthy persons. A significant

**Table 2. Percentage variations in Liver enzymes, Bilirubin, and Kidney function tests in healthy individuals, methotrexate and leflunomide treated rheumatoid arthritis patients.**

| Groups | Variables | Mean + SEM | Percentage variation | P value |
|---|---|---|---|---|
| | | **Liver Function Tests** | | |
| **Control** | ALT | 17.54 ± 2.14 | | |
| | AST | 26.36 ± 1.91 | | |
| | ALP | 262.0 ± 18.33 | | |
| | Total Bilirubin | 0.81 ± 0.03 | | |
| | Indirect Bilirubin | 0.36 ± 0.02 | | |
| | Direct Bilirubin | 0.49 ± 0.02 | | |
| **MTX treated** | ALT | 42.68 ± 2.14 | 145.28% (increase) | <0.0001 |
| | AST | 41.86 ± 1.91 | 58.80% (increase) | <0.0001 |
| | ALP | 289.4 ± 18.33 | 10.45% (increase) | NS |
| | Total Bilirubin | 0.82 ± 0.03 | 1.87% (increase) | 0.8 NS |
| | Indirect Bilirubin | 0.41 ± 0.02 | 14.4 (increase) | 0.01 |
| | Direct Bilirubin | 0.41 ± 0.02 | 14.7 (decrease) | 0.03 |
| **LEF treated** | ALT | 43.14 ± 2.14 | 147.81% (increase) | <0.0001 |
| | AST | 43.52 ± 1.91 | 65.09% (increase) | <0.0001 |
| | ALP | 312.1 ± 18.33 | 19.12% (increase) | 0.02 |
| | Total Bilirubin | 0.82 ± 0.03 | 1.47% (increase) | 0.8 NS |
| | Indirect Bilirubin | 0.34 ± 0.02 | 3.8 (decrease) | 0.01 |
| | Direct Bilirubin | 0.45 ± 0.02 | 8.13 (decrease) | 0.03 |
| | | **Renal Function Test** | | |
| **Control** | BUN | 19.36 ± 3.92 | | |
| | Serum creatinine | 0.82 ± 0.19 | | |
| | GFR | 100.6 ± 5.50 | | |
| | Urinary Uric Acid | 6.36 ± 0.17 | | |
| | Urinary Urea | 15.98 ± 0.25 | | |
| **MTX treated** | BUN | 34.20 ± 3.92 | 76.65 (increase) | 0.001 |
| | Serum creatinine | 1.55 ± 0.19 | 88% (Increase) | 0.0007 |
| | GFR | 70.76 ± 5.50 | 29.6% (decrease) | <0.0001 |
| | Urinary Uric Acid | 2.86 ± 0.17 | 55.03% (decrease) | <0.0001 |
| | Urinary Urea | 5.55 ± 0.25 | 65% (decrease) | <0.0001 |
| **LEF treated** | BUN | 26.94 ± 3.92 | 39.15 (increase) | 0.001 |
| | Serum creatinine | 1.012 ± 0.19 | 22.8% (increase) | 0.05 |
| | GFR | 79.73 ± 5.50 | 20.7% (decrease) | <0.0001 |
| | Urinary Uric Acid | 2.94 ± 0.17 | 53.77% (decrease) | <0.0001 |
| | Urinary Urea | 5.96 ± 0.25 | 62% (decrease) | <0.0001 |

reduction in GFR was also noticed by leflunomide usage but this reduction was 12% less than MTX administered group.

Persistent use of DMARDs reduced the urinary uric acid up to ≥3 folds. The mean urinary uric acid value in MTX and LEF users was found to be 2.86 ± 0.17 mg/dL and 2.94 ± 0.17 mg/dL, respectively. That was 55% and 54% less than the urinary uric acid of healthy subjects. These finding indicated substantial uric acid reduction in DMARD treated patients. Both type of drugs depicted similar effect with negligible difference in their mean values.

Analogous pattern of reduction was also observed in urinary urea of MTX and LEF treated patients as compared to healthy individuals. The mean value of urinary urea was 15.98 ± 0.25 mg/dL in healthy individuals which decreased to 5.96 ± 0.25 mg/dL in LEF users and 5.55 ± 0.25 mg/dL in MTX users. The overall reduction was 65% and 62% in MTX and LEF treated groups, nearly three times less than healthy group (Table 2 and Fig 6A–6E).

## Discussion

Incidence of rheumatoid arthritis is reported more in obese and short stature people. Obesity is strongly correlated with chronic inflammatory diseases and females are more susceptible to

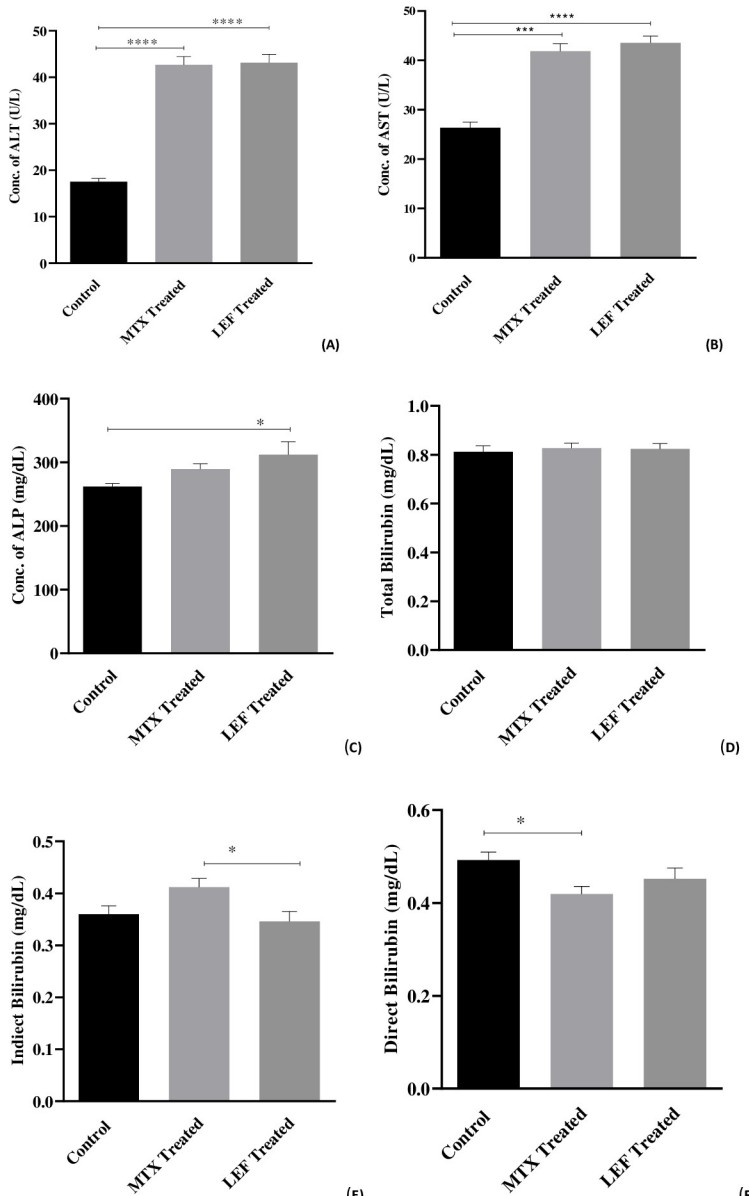

**Fig 5.** (A-F): Average concentrations of AST, ALP, ALT, Total bilirubin, direct bilirubin (E) and indirect bilirubin. *, **, *** indicate significance at P< 0.05, 0.01 and P< 0.001.

inflammation because of hormones. Obesity, along with inflammation and oxidative stress, sets an alarming signal for rheumatoid arthritis [13,14].

Duration of drug use is an important determinant of its beneficial effects. DMRDS are immune suppressers and slow down the progression of RA but the persistent use of DMARDs may develop toxicity in various organs. The current study revealed that long term use of DMARDs (methotrexate and leflunomide or combination) significantly increased the blood pressure and amplify the risk of cardiovascular diseases. Long term exposure of MTX may play a significant role in determining the beneficial cardiovascular effects of the drug and persistent use of methotrexate is linked to a lower incidence of CVD-related events in RA patients, thus reducing the burden of CVD in rheumatoid arthritis [15]. Reducing the severity of

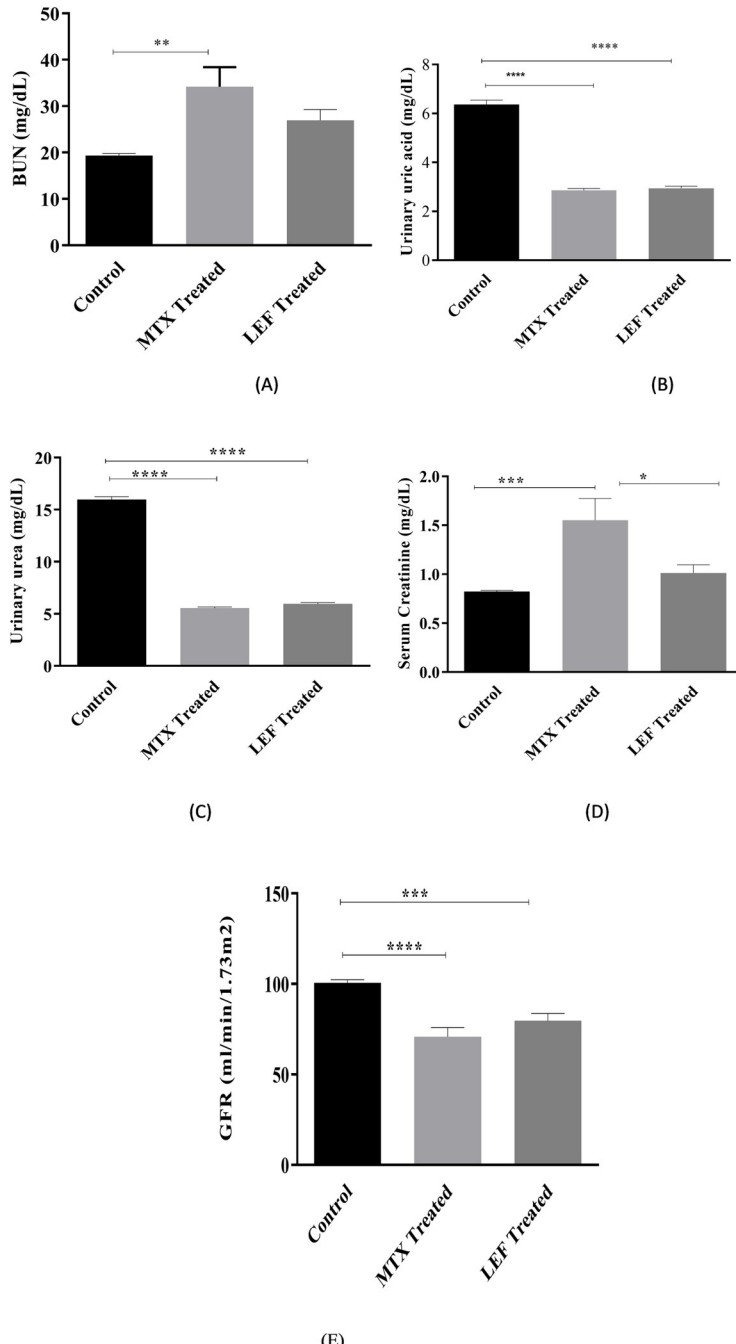

**Fig 6.** (A-E): Average levels of Blood Urea Nitrogen (BUN), uric acid, urinary urea, serum creatinine, and GFR. *, **, ***indicate significance at P<0.05, 0.01 and P<0.001.

inflammation with MTX may lessen collateral damage like atherosclerosis while simultaneously improving disease-specific outcomes.

Chronic use of DMARDs also causes liver toxicity and liver dysfunction. Different sorts of liver issues may be indicated by bilirubin levels that are higher than usual. Bilirubin is a protective factor for RA and the development of RA has an inverse relationship with the serum total bilirubin. This protective effect of bilirubin was linked to its physiologic anti-inflammatory

effects. Thus, lower bilirubin level indicated the disease severity. The findings confirmed that MTX is more toxic for liver than LEF. Normally, the amount of ALT in the blood is minimal, but when the liver is injured, or heart is at risk of atherosclerosis, ALT is released into the blood and its level rises [16]. The present study determined that both drugs are equally toxic to liver as negligible difference was observed in their mean values. Methotrexate treatment for an extended period has been associated with elevated levels of serum AST and the onset of cirrhosis, fibrosis, and fatty liver disease. Levels of serum ALP are typically increased in people with rheumatoid arthritis who also have a range of liver disorders, such as stenosis of extra hepatic bile duct, intra hepatic cholestasis, infiltrative liver disease, and hepatitis [17]. The present study concluded that LEF treatment causes more elevation in AST and ALT than MTX. This finding is supported by Whitelock *et al.* (2009) and Di Martino *et al.* (2023) who reported that elevations in liver function tests, particularly in ALT and AST, were linked to leflunomide administration [18,19].

Compatible to earlier studies, we found that renal involvement in RA patients is common and that the most important factors affecting renal function in these individuals are age and drug used. Age and kidney function should be interpreted with caution because people lose nephrons and renal function as they aged. In the current study, age may not explain renal failure because the average age of our patients was about 45 years, which is considered a young population by the WHO age group categorization [20,21]. Thus, the severity of disease and used drug are the important determinant of renal impairment. The increased amount of BUN by the persistent use of MTX indicates the degradation of RBCs that is the most dangerous toxic effect of MTX on liver as describe earlier. Similarly, MTX also raised the serum creatinine level more than 88% and exhibit high BUN to serum creatinine ratio. Here again LEF found comparatively less toxic and increase in creatinine level is 22.8% than control. Higher serum creatinine reduced the GFR, and highly significant reduction was noticed by MTX. on the contrary, MTX lowered the level of urinary uric acid and urinary urea, resulting in the less inflammation and disease severity. Consequently, MTX may be helpful for soothing the pains of RA but highly dangerous for kidneys.

## Conclusion

Rheumatoid arthritis is an autoimmune disease that reported in all adult age group. Harmful effects of anti-rheumatic drugs are more than their benefits and multiorgan toxicity is reported by the long-term use of MTX and LEF. Methotrexate was prescribed to youngster with early stage of disease while leflunomide was suitable at advanced stage of disease. Patients treated with methotrexate put on more weight than leflunomide treated patients. The risk of getting hepatotoxicity is greater by long term use of LEF but it was found less harmful for respiratory system and cardiac output. Methotrexate, on the other hand, is more toxic to kidneys and cardiovascular system resulting in increased blood pressure. However, it reduces the inflammation by decreasing the uric acid and urea. Thus, it can be concluded that both drugs are interchangeable and may prescribed alternately for lesser side effects.

## Supporting information

**S1 Fig. The lay out of patient enrollment in the study based on inclusion and exclusion criteria.**
(TIF)

**S1 File. Questionnaire.**
(PDF)

**S1 Table. Minimal data set of healthy control group.**
(PDF)

**S2 Table. Minimal data set of rheumatoid arthritis patient treated with methotrexate.**
(PDF)

**S3 Table. Minimal data set of rheumatoid arthritis patient treated with leflunomide.**
(PDF)

## Acknowledgments

Authors are much obliged to Dr. Atta-ur-Rehman and Dr. Faisal Ejaz, rheumatologist at medical unit 1 Rheumatology Department, Allied Hospital, Faisalabad for their support in getting rheumatoid arthritis patients. Authors are also indebted to lab technicians Mr. Ahtasham and Mr. Abdullah for their cooperation in lab tests performance.

## Author Contributions

**Conceptualization:** Tamseela Mumtaz.

**Data curation:** Ayesha Tahir, Maham Almas Tariq.

**Formal analysis:** Ayesha Tahir, Maham Almas Tariq.

**Investigation:** Ayesha Tahir, Maham Almas Tariq.

**Methodology:** Tamseela Mumtaz.

**Project administration:** Tamseela Mumtaz.

**Resources:** Tamseela Mumtaz.

**Supervision:** Tamseela Mumtaz.

**Writing – original draft:** Tamseela Mumtaz.

**Writing – review & editing:** Tamseela Mumtaz, Muhammad Amir Iqbal.

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
