## [Decision Letter · Decision Letter 0]

28 May 2023

PONE-D-23-10597Comparative Analysis of Multiorgan Toxicity Induced by Long Term Use of Disease Modifying Anti-Rheumatic DrugsPLOS ONE

Dear Dr. Mumtaz,

Thank you for submitting your manuscript to PLOS ONE. After careful consideration, we feel that it has merit but does not fully meet PLOS ONE’s publication criteria as it currently stands. Therefore, we invite you to submit a revised version of the manuscript that addresses the points raised during the review process.

Revise the whole manuscript keeping in view the comments and suggestions of both reviewers. Submit the revised version of the article. 

We look forward to receiving your revised manuscript.

Kind regards,

Samiullah Khan, Ph. D

Academic Editor

PLOS ONE

Journal Requirements:

4. Please upload a new copy of Figures 5 and 6 as the detail is not clear. Please follow the link for more information: https://blogs.plos.org/plos/2019/06/looking-good-tips-for-creating-your-plos-figures-graphics/

Additional Editor Comments:

Dear author,

Revise carefully the whole manuscript keeping in view the comments and suggestions of both reviewers. The revised version of article should be submitted accordingly.

Thanks

Reviewers' comments:

Reviewer's Responses to Questions

**Comments to the Author**

1. Is the manuscript technically sound, and do the data support the conclusions?

Reviewer #1: Partly

Reviewer #2: Yes

2. Has the statistical analysis been performed appropriately and rigorously? 

Reviewer #1: No

Reviewer #2: Yes

3. Have the authors made all data underlying the findings in their manuscript fully available?

Reviewer #1: No

Reviewer #2: Yes

4. Is the manuscript presented in an intelligible fashion and written in standard English?

Reviewer #1: Yes

Reviewer #2: Yes

5. Review Comments to the Author

Reviewer #1: Title was “Comparative Analysis of Multiorgan Toxicity Induced by Long Term Use of Disease Modifying Anti-Rheumatic Drugs”, ideally title should Indicate the study’s design with a commonly used term in the title or the abstract

Study objective is not properly specified, state specific objectives, including any pre-specified hypotheses. There is no clear statement of objective, its just mentioned “Keeping in mind the above-mentioned points, this comprehensively designed research was carried out to sort out the lowest destructive drug with great benefits for treatment of RA”

The title, objective and rationale of the study are not aligned. Rationale states “The risk to benefit ration of MTX and LEF by comparing the efficacy and toxicity of these drugs”.while title was “Comparative Analysis of Multiorgan Toxicity Induced by Long Term Use of Disease Modifying Anti-Rheumatic Drugs”.

The literature says “if data are available at no extra cost, then we may recruit multiple controls for each case. However, if it is expensive to collect exposure and outcome information from cases and controls, then the optimal ratio is 4 controls: 1 case”. On the contrary in this study number of controls was half as compared to a number of cases.

Matching is often used in case-control control studies to ensure that the cases and controls are similar in certain characteristics. However there was no matching done as it is mentioned that “50 subjects from random population as control”. Control must be matched for certain variables like sex and age while in this case no matching was done and controls were selected randomy from the general population .

The eligibility criteria, and the sources and methods of case ascertainment and control selection is not clear . There were three groups in total and matching is not group wise that makes the validity of results doubtful. The rationale for the choice of cases and controls in two groups is ambiguous . DMARDS, Methotrexate (MTX) and Leflunomide (LEF) is used interchangeably . The comparability of assessment methods is not mentioned in more than than one group. The outcomes, exposures, predictors, potential confounders, and effect modifiers are not catered for .

There is no explanation about how the study size was arrived. Data handling of quantitative variables is not explained properly. Statistical analysis lacks an explanation of how confounding subgroups and interactions are handled ,how missing data were addressed. how matching of cases and controls was addressed any sensitivity analyses. Its better to report numbers of individuals at each stage of study eg numbers potentially eligible, examined for eligibility, confirmed eligible, included in the study, completing follow-up, and analysed, reasons for non-participation at each stage, consider use of a flow diagram to make it clear and explicit . In confounder-adjusted estimates and their precision (eg, 95% confidence interval) is not explained. Percentage variation is not done for control group, justification for using Mean , SEM and significance of statistical difference is not clear . Conclusion do not supports overall interpretation of results considering objectives, limitations, results from similar studies, and other relevant evidence.

Reviewer #2: Dear Editor,

The article " Comparative Analysis of Multiorgan Toxicity Induced by Long Term Use of Disease

Modifying Anti-Rheumatic Drugs" was reviewed. It is a good study. The authors describe the potential of both drugs prescribed as treatment of arthritis. I recommend the publication of manuscript after some revision.

The authors have done the good research work and their work is appreciable. But I find some of the points according to my knowledge.

I have appended below some comments that the authors may wish to consider:

it has been attached as a file review report.

6. PLOS authors have the option to publish the peer review history of their article (what does this mean?). If published, this will include your full peer review and any attached files.

Reviewer #1: **Yes: **Khola Noreen

Reviewer #2: No

---

## [Author Response · Author response to Decision Letter 0]

8 Jun 2023

All the authors are very thankful for considering the manuscript for peer review. We are much obliged for valuable comments and suggestions of academic editor and reviewers to improve the quality of the article. The manuscript has been revised accordingly. A detailed, point to point response to the academic editor and reviewers’ comments is as follow

Response to Academic Editor

1. There is no change in financial disclosure statement. This research work did not receive any grant from funding agency in the public, commercial or industrial sector.

2. The protocols used in the study are recommended by IFCC (International Federation of Clinical Chemistry and Laboratory Medicine) through commercially available kits developed by DiaSys Diagnostic Systems GmbH Germany. The made and catalogue numbers of all kits are mentioned in revised manuscript. We are agreed to your recommendation of submitting laboratory protocols in protocols.io to enhance the reproducibility of results. We will definitely work on it.

Response to Journal Requirements

PLOS ONE stye requirements

1. The manuscript has been revised according to PLOS ONE formatting guidelines for main body of manuscript, authors affiliations and file naming. The authors have tried our level best to make the article as per journal style requirement but still man is prone to error, even if a mistake is made, we are ready to correct it.

2. The manuscript has been thoroughly checked and proof read for language use, spelling and grammar by the authors themselves and no professional services for copyedit is used.

A copy of manuscript showing changes using “track changes” has been uploaded as a *supporting information* file. Revised version of manuscript without track changes named as “manuscript” has been uploaded

3. We apologize for misunderstanding the purpose of data availability statement. All the study’s minimal data sets with anonymous patient ID have been uploaded as supporting information files. 

4. We feel sorry for the poor quality of figures and are much thankful to have PACE – the Picture Analysis and Conversion Engine (https://pacev2.apexcovantage.com). The application has made it very easy to convert image into required format. In fact, I learned a lot while revising the manuscript for PLOS ONE 

Response to reviewer 1(Khola Noureen)

The authors are much obliged for such an in-depth review. In deed it was a very comprehensive review and all aspect of the study were observed with eagle eye. This detailed review helped us a lot to make the article worth writing. Following are satisfactory answers of the comments 

1. Title was “Comparative Analysis of Multiorgan Toxicity Induced by Long Term Use of Disease Modifying Anti-Rheumatic Drugs”, ideally title should Indicate the study’s design with a commonly used term in the title or the abstract

As for as title is concerned, it has no space for study design. Generally, study design is defined in abstract and is mentioned in abstract (page 2, line 3-4) 

2. Study objective is not properly specified, state specific objectives, including any pre-specified hypotheses.

In deed it is a valid point. Study objectives are thoroughly considered and revised as per suggestion. Specifically. Underlying hypothesis and specific objectives are included in Abstract “objective section” (page 2 line 2-5)

3. The title, objective and rationale of the study are not aligned. 

The title, objective and rationale of study are revised and integrated to each other (page 4 line 59-66)

4. Number of controls was half as compared to a number of cases

The ratio of case to control is1:1. Actually the number of participants in each group was equal. The control subjects were same for each case group (Fifty healthy subjects were recruited against fifty (50) RA patients treated with MTX and the same healthy subjects were compared against RA patients treated with LEF. There were two basic reasons for same control group

a. The healthy individuals were fully screened for complete medical and physical fitness and subsequently found to be perfectly healthy subjects 

b. Since the study was not funded by anywhere, we compared the same control for both groups considering the financial burden.

5. Control must be matched for certain variables like sex and age

Revised as suggested (page 5 Line 87-88)

6. The eligibility criteria, and the sources and methods of case ascertainment and control selection is not clear, how matching of cases and controls was addressed any sensitivity analyses. It is better to report numbers of individuals at each stage of study e.g. numbers potentially eligible, examined for eligibility, confirmed eligible, included in the study, completing follow-up, and analyzed, reasons for non-participation at each stage, consider use of a flow diagram to make it clear and explicit.

Revised. See Supporting Information S1 Fig (page 5 line 81)

7. There is no explanation about how the study size was arrived

Explained on Page 5 line 86-87

8. Data handling of quantitative variables is not explained properly. Statistical analysis lacks an explanation of how confounding subgroups and interactions are handled; how missing data were addressed. In confounder-adjusted estimates and their precision (eg, 95% confidence interval) is not explained. 

STROBE check list is attached as other supporting document

9. Percentage variation is not done for control group, justification for using Mean, SEM and significance of statistical difference is not clear

Mean values of each case group were compared with mean value of control group to calculate the percentage difference from control group. As mean value of control group was used as reference, percentage variation cannot be done for standard or reference value. 

10. Conclusion do not support overall interpretation of results considering objectives, limitations, results from similar studies, and other relevant evidence.

Conclusion has been revised in light of objectives, study limitations and results (page 16, line 295-304). 

Response to Reviewer 2

The authors are very much grateful for your positive reviews. Your positive approach and encouragement have shown us a new path. We are happy to implement your valuable suggestion and feel pleasure to answers your kind questions. The points are as follows 

1. Page 2: Abstract: Under Methodology ‘‘Stress on liver was estimated by measuring

change in liver enzymes (ALT, AST, and ALP) and total bilirubin. While, degree of renal

impairment was assessed by calculating GFR’’ the abbreviations are not preferred in the

abstract so author should be mentioned the complete names.

Incorporated as suggested (page 2 line 12-14)

2. Page 3: Introduction: In the last paragraph ‘’ First, females have stronger and more reactive immune system than men, secondly, feminine hormones also flare RA risks’’. These reasons not clear the author claim in the previous lines. It should be clearer.

Statement made more clearer (page 3 line 43-45) 

3. Page 5: Materials and methods: the author has claimed the teen agers not included in the study. It is better to mention the age limit also.

Age was already mentioned on page 5 line 96. However, for more convenient reading, the word teenager replaced with “participants of less than 18 years” (Page 5 line 84)

4. Page 7: Result: same reservation about age. It is mentioned middle age and last decade of age which is not clear the proper age limit.

Age range has been mentioned (Page 7 line 129-131)

5. Page 8: it is stated that ‘‘LEF treatment also speeds up the pulse rate (10.8%) and this elevation is statistically significant (P=0.0002)’’. It may be corrected as (p<0.0002). It should be properly checked according to the value mentioned in the table.

Very good observation. Authors appreciated the keen interest of reviewer in result interpretation. It was really a mistake. Results has been carefully checked and revised accordingly (page 8 line 158, page 9 Table 1) 

6. Page 13: Discussion section 2nd paragraph. The author discusses about the problem with the age 53 included in the study where as they have not mentioned clearly the age limits in the materials, so more explanation needed here

The average age of study participant was 45 years. It has been mentioned in Results section under heading “age distribution” on page 7 line 130. Correction in average age has been made on page 15 line 283. 

7. References: All the references mentioned in the references list are not according to the instructions provided in the author guidelines.

References are formatted according to guideline.

8. The overall format of the manuscript needs revision according to the author guide line provided by the PLOS One: abstract should be in paragraph form. Format of the headings also needs revision. Reference style also needs revision: full Journal names are mentioned in the references it should be abbreviated. 

The whole manuscript has been revised and prepared according to PLOS ONE guidelines 

We believe that manuscript is now suitable for publication in PLOS ONE

---

## [Decision Letter · Decision Letter 1]

4 Jul 2023

PONE-D-23-10597R1Comparative Analysis of Multiorgan Toxicity Induced by Long Term Use of Disease Modifying Anti-Rheumatic DrugsPLOS ONE

Dear Dr. Mumtaz, 

Thank you for submitting your manuscript to PLOS ONE. After careful consideration, we feel that it has merit but does not fully meet PLOS ONE’s publication criteria as it currently stands. Therefore, we invite you to submit a revised version of the manuscript that addresses the points raised during the review process.

We look forward to receiving your revised manuscript.

Kind regards,

Samiullah Khan, Ph. D

Academic Editor

PLOS ONE

Journal Requirements:

**Additional Editor Comments:**

Dear Author,

The manuscript has been thoroughly revised but reviewer 1 suggested some valid minor corrections. The author should re-revise the manuscript and incorporate the reviewer's suggestion. The author should also include some recent relevant (2021-2023) references in the introduction as well as in the discussion section.

Thanks

Reviewers' comments:

Reviewer's Responses to Questions

**Comments to the Author**

1. If the authors have adequately addressed your comments raised in a previous round of review and you feel that this manuscript is now acceptable for publication, you may indicate that here to bypass the “Comments to the Author” section, enter your conflict of interest statement in the “Confidential to Editor” section, and submit your "Accept" recommendation.

Reviewer #1: (No Response)

Reviewer #2: All comments have been addressed

2. Is the manuscript technically sound, and do the data support the conclusions?

Reviewer #1: Yes

Reviewer #2: Yes

3. Has the statistical analysis been performed appropriately and rigorously? 

Reviewer #1: Yes

Reviewer #2: Yes

4. Have the authors made all data underlying the findings in their manuscript fully available?

Reviewer #1: Yes

Reviewer #2: Yes

5. Is the manuscript presented in an intelligible fashion and written in standard English?

Reviewer #1: Yes

Reviewer #2: Yes

6. Review Comments to the Author

Reviewer #1: Objective: The constant use of disease modifying anti rheumatic drugs affects the functioning

of multiple organs inside the body. Some drugs are more toxic than others. The present case

control investigation was designed to evaluate the comparative toxicity of methotrexate and

leflunomide on multiple organs. in rheumatoid arthritis patients. For this purpose, 100 subjects ..............

There are few points to be consider in abstract section;

1. Reviewer wrote all of the abstract under heading of objective ....statement starting from for this purpose ....is part of methodology rather than objective...

2. Abstract is unstructured , all has been written under one heading heading

3. in abstract its mentioned "Participants of the study were categorized into three groups,1) Rheumatoid arthritis patients on methotrexate treatment, 2) Rheumatoid arthritis patients on leflunomide treatment 3) healthy subjects". (Number of participants allocated to each group must be mentioned ...there should be a clear description as how 100 cases and 50 controls were allocated into different group specially mentioning the number of each group , since its a case control study , this information is very pertinent as ratio of allocation of cases to control and for matched studies, its important to give matching criteria and the number of controls per case which is not clear even in revised manuscript .

In methodology section its mentioned " This is a case control study comprises hundred (100) subjects diagnosed with rheumatoid arthritis and fifty (50) age and sex matched healthy subjects from random population as control." even this does not describe methods of case ascertainment into each group.

4. In line 94, 95 its mentioned "chosen rheumatoid arthritis subjects were categorized into two subgroups based on oral administration of two different types of DMARDs (Group 1: methotrexate (n=50), Group 2: leflunomide (n=50) "however this statement has no mention of allocation of controls into each group.

Sample size estimation is still not done as per statistical rule , sample size calculation in statistical terms is based on determination of the required number of cases and controls is based on consideration of the strength of the relationship between the disease and exposure to the putative cause, the variability in exposure within the population under study, and the desired size and power of the statistical test, which is still missing .

Issues raised in bullet point 8 & 9 are still not described

Results: under heading of result ..line 127 , its mentioned "Overall, 80% patients (n=80) were treated with comparatively low dose (10 mg)" author should mention the potential confounding effect of variation in doses

side effects mentioned in line 138 "Both methotrexate and leflunomide had some side effects such as fatigue,

139 headache, alopecia, teeth pain, rash, dizziness, sleep problem, cough, depression, breathing

140 problem and muscle stretching" are quite vague , author should operationally define these terms and explicitly mention how these complaints were measured

in line 174 and 175 " patients have high heart rates that may be attributed to other co morbidities like obesity, insulin resistance etc" these are potential confounders which need to be control during participants recruitment phase

Reviewer #2: (No Response)

7. PLOS authors have the option to publish the peer review history of their article (what does this mean?). If published, this will include your full peer review and any attached files.

Reviewer #1: **Yes: **Dr Khola Noreen

Reviewer #2: No

---

## [Author Response · Author response to Decision Letter 1]

19 Jul 2023

My co-authors and I were pleased to receive your response on 5th July 2023 inviting us for minor revision and resubmission of article entitled “Comparative analysis of multiorgan toxicity induced by long term use of disease modifying anti-rheumatic drugs” (manuscript ID PONE-D-23-10597R1). We are very much obliged for your precious time and efforts in reviewing our manuscript. Your quick comments are very helpful in improving the contents and presentations of our manuscript. Accordingly, we added the recent references to make the article up to date.

On behalf of all my co-authors, I would like to extend my gratitude for positive and constructive remarks of both reviewers. The comments of reviewer 1 are valid and depicted the reviewer strong epidemiological/statistical background. The points raised on sample size and sample recruitment/selection are very logical and draw our attention to a very important point which we forgot to address. After careful response to every point, we would like to submit the revised manuscript for reconsideration for publication in PLOS One Journal. 

We also thanks to reviewer 2 for recommended the article for publication

We believe that manuscript is now suitable for publication in PLOS ONE

The recommendations and advices help us a lot in improving the quality of the article. We have revised the manuscript according to the reviewer comments and explained all the points under “response”. The tracked manuscript also showed minor revision and added/ renumbered reference.

---

## [Editor Report · Decision Letter 2]

13 Aug 2023

Comparative Analysis of Multiorgan Toxicity Induced by Long Term Use of Disease Modifying Anti-Rheumatic Drugs

PONE-D-23-10597R2

Dear Dr. Tamseela,

We’re pleased to inform you that your manuscript has been judged scientifically suitable for publication and will be formally accepted for publication once it meets all outstanding technical requirements.

Kind regards,

Samiullah Khan, Ph. D

Academic Editor

PLOS ONE
---

## [Editor Report · Acceptance letter]

17 Aug 2023

PONE-D-23-10597R2 

Comparative analysis of multiorgan toxicity induced by long term use of disease modifying anti-rheumatic drugs 

Dear Dr. Mumtaz:

I'm pleased to inform you that your manuscript has been deemed suitable for publication in PLOS ONE. Congratulations! Your manuscript is now with our production department. 

Kind regards, 

on behalf of

Dr. Samiullah Khan 

Academic Editor

PLOS ONE